# Discussing prognosis and the end of life with patients with advanced cancer or COPD: A qualitative study

**Catherine Owusuaa** [1] *, **Liza G. G. van Lent** [1], **Adriaan van 't Spijker** [2], **Carin C. D. van der Rijt** [1], **Agnes van der Heide** [3]

**1** Department of Medical Oncology, Erasmus MC Cancer Institute, Rotterdam, The Netherlands,
**2** Department of Medical Psychology, Erasmus MC, Erasmus University Medical Center, Rotterdam, The Netherlands, **3** Department of Public Health, Erasmus MC, Erasmus University Medical Center, Rotterdam, The Netherlands

* c.owusuaa@erasmusmc.nl

**Data Availability Statement:** The summary of the interview transcripts in Dutch are available from the corresponding author on reasonable request. The data are restricted, because the original transcripts contain potentially identifying or sensitive patient

## Abstract

### Objectives

To explore patients' experiences and recommendations for discussions about their prognosis and end of life with their physicians.

### Methods

Patients with advanced cancer or advanced chronic obstructive pulmonary disease (COPD) were enrolled in qualitative interviews, which were analyzed with a phenomenological and thematic approach.

### Results

During interviews with fourteen patients (median age 64 years), we identified the following themes for discussion about prognosis and the end of life: topics discussed, the timing, the setting, physician–patient relationship, responsibilities for clinicians, and recommendations. Patients preferred the physician to initiate such discussion, but wanted to decide about its continuation and content. The discussions were facilitated by an established physician–patient relationship or attendance of relatives. Patients with cancer had had discussions about prognosis at rather clear-cut moments of deterioration than patients with COPD. Patients with COPD did not consider end-of-life discussions a responsibility of the pulmonologist. Patients recommended an understandable message, involvement of relatives or other clinicians, sufficient time, and sensitive non-verbal communication.

### Conclusions

Patients appreciated open, sensitive, and negotiable discussions about prognosis and the end of life.

information. The restriction on the original transcripts is imposed by the Medical Ethical Research Committee of the Erasmus University Medical Center (MEC-2017-289 addendum). The Research Support Office of the Public Health department at Erasmus MC is able to field data access queries. The Research Support Office can be contacted using the email address: rso.mgz@erasmusmc.nl. To request the data set, please refer to Panama-number 3366.

**Funding:** C.C.D.vd.R acquired the funding for this study, which was by a grant from the Netherlands Organization for Health Research and Development (ZonMw; https://www.zonmw.nl/en/; grant number 844001209). The funder had no role in study design, data collection and analysis, decision to publish, or preparation of the manuscript.

**Competing interests:** The authors have declared that no competing interests exist.

## Practice implications

Patients' recommendations could be used for communication training. Possible differences in the need for such discussions between patients with cancer or COPD warrant further research.

## 1. Introduction

Adequate prognostic information can facilitate the initiation of end-of-life discussions or advance care planning, which is aimed at increasing the concordance between preferred and delivered end of life care, especially in the last year of life [1–6]. Patients' understanding of their prognosis can support them in evaluating their health situation and reflecting on their treatment preferences [5,6]. However, discussions of patients' prognosis and end of life should adequately fit in patients' experience of their illness trajectory.

Different illness trajectories at the end of life have been identified. A period of rapid deterioration shortly before death is common for advanced cancer, whereas a trajectory with gradual decline and recurrent life-threatening exacerbations is common in patients with chronic organ failure, with advanced chronic obstructive pulmonary disease (COPD) being a typical example [7]. Patients' beliefs about their illness trajectory and prognosis may affect their willingness to reflect on and discuss end of life. Multiple studies have concluded that most patients with cancer appreciate a discussion of their prognosis with their physicians [8,9]. A study by Curtis et al. found that most oxygen-dependent COPD patients rather focused on staying alive than on talking about death or a potentially poor prognosis [8]. Furthermore, in a study by Fried et al., 83% of patients with cancer, COPD, or congestive heart failure who believed they had one year or less to live, desired prognostic information, whereas only 50% of patients who believed they had longer than five years to live, had that desire [10].

Despite these seemingly different preferences of patients with different illness trajectories, little is known about their actual experience of prognostic or end-of-life discussions. The aim of our study was therefore to explore the experiences of patients with advanced cancer and patients with advanced COPD regarding those discussions, as well as their preferences and recommendations for such discussions.

## 2. Methods

### 2.1 Patients

Medical specialists from one academic (Erasmus MC) and two teaching (Amphia and Maasstad Hospital) hospitals in the Netherlands were asked to invite patients for an in-depth interview between March and June 2018. Patients were eligible if they had advanced cancer, i.e. were treated with palliative intent, or advanced COPD, i.e. with severe or very severe airflow limitation according to the Global Initiative for Chronic Obstructive Lung Disease classification (Gold III or IV); were likely to die within a year; and had recently discussed their prognosis and end of life with their medical specialist [11]. Patients who gave permission to be contacted by a researcher (CO) were asked to participate in a 1-hour interview. Patients who felt too ill to partake in an interview were excluded.

### 2.2 Patient and public involvement

We used a semi-structured interview guide to discuss patients' experiences and preferences, which was developed in consultation with a representative of a patient and relative advisory

board (Box 1). The interview guide was tested in a pilot interview with a relative of a deceased (cancer) patient. Afterwards, the interview guide was reevaluated with the representative.

---

### Box 1. Interview guide

Tell me about your illness trajectory from diagnosis until now.

Have you discussed your life expectancy (prognosis) with your physician? In what way?

Have you discussed end of life topics/issues with your physician? In what way?

Tell me about this conversation:

When was it?

Where was it?

Which topics were discussed?

Who was present?

What was your general experience of the conversation?

What did you find pleasant about the conversation?

What did you find less pleasant about the conversation?

What did you find pleasant about the communication of the physician?

What did you find less pleasant about the communication of the physician?

How is your relationship with the medical specialist?

How is your relationship with the general practitioner?

Do you have any general recommendations for physicians to improve the communication?

---

## 2.3 Procedures

The Medical Ethical Research Committee of the Erasmus University Medical Center declared that there were no objections to performing this study (MEC-2017-289 addendum). One researcher (CO) interviewed all patients at a location of their choice after obtaining their written informed consent. The researcher took field notes during the interviews. Interviews were conducted until the research team agreed that data saturation was reached, i.e. until no new information surfaced.

## 2.4 Design and analysis

All interviews were audio recorded, transcribed verbatim and initially analyzed by two researchers with a background in medicine (CO) and communication (LGGvL). They read a set of three transcripts independently, using a phenomenological interpretive approach [12]. With this approach we aimed to examine the experiences and preferences of the patients based on their own descriptions and phrases. From those descriptions, we derived and categorized (sub)themes. We applied those (sub)themes to all interview transcripts and performed a thematic analysis [13]. The final (sub)themes were summarized in a coding tree (Fig 1). Finally,

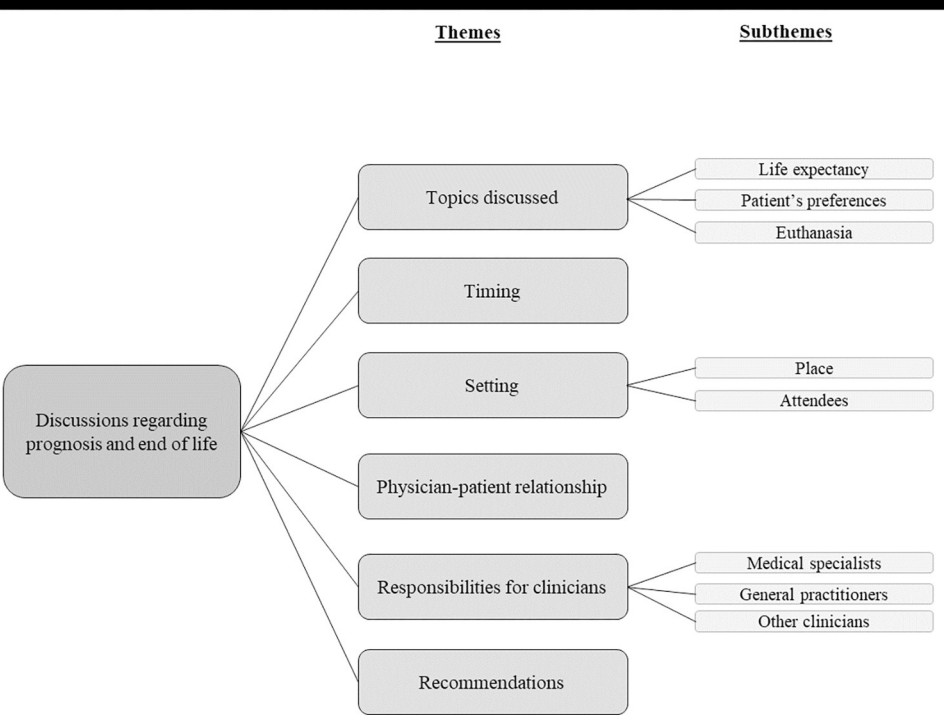

**Fig 1. Coding tree.**

CO selected relevant quotes per theme, which were approved by all authors. We used the Standards for Reporting Qualitative Research guideline to structure the manuscript [14].

We continuously evaluated and reflected on the processes of the study. First, CO and LGGvL independently derived (sub)themes from the initial three transcripts, which they evaluated together. Second, the researchers categorized the (sub)themes and applied them to the other transcripts. Third, the (sub)themes were discussed with CCDvdR and AvdH. Discrepancies were resolved by discussions between all researchers.

## 3. Results

Fourteen patients (six male and eight female patients) out of sixteen who were contacted could be interviewed; one patient with cancer felt too ill and another patient with cancer did not consent to recording. The participating patients were invited by six oncologists and two pulmonologists. Patients had a median age of 64 years (32–80) and half of them had cancer as their main diagnosis (Table 1). Eleven patients were interviewed at their house and two patients at the hospital during a visit at the outpatient clinic. Five patients were accompanied by a relative during the interview.

We identified the following main themes in our analysis of the interview transcripts: topics discussed, timing, setting, physician–patient relationship, responsibilities for clinicians, and recommendations (Fig 1). These themes will be discussed consecutively.

### 3.1 Topics discussed

**Life expectancy.** Patients with cancer were provided with relatively cautious estimates of their remaining life expectancy, which was expressed in e.g. 'months' or 'years'. Such information was generally supported by statistics on antitumor treatments. Some patients experienced

**Table 1. Patients' characteristics.**

|  | Patients (%) N = 14 |
|---|---|
| Hospital | |
| Erasmus MC | 5 (36) |
| Amphia | 6 (43) |
| Maasstad Hospital | 3 (21) |
| Age, median (interquartile range) | 64 (32–80) |
| Sex | |
| Male | 6 (43) |
| Female | 8 (57) |
| Race/ethnicity | |
| Caucasian | 14 (100) |
| Primary diagnosis | |
| Cancer | 7 (50) |
| COPD | 7 (50) |
| Type of cancer (n = 7) | |
| Colorectal | 4 (57) |
| Prostate | 1 (14) |
| Ovary | 1 (14) |
| Pancreatic | 1 (14) |
| COPD classification (n = 7) | |
| Gold III | 2 (29) |
| Gold IV | 5 (71) |
| Setting | |
| Home | 11 (79) |
| Hospital | 2 (14) |
| Workplace | 1 (7) |
| Accompanied by a relative | |
| No | 9 (64) |
| Yes | 5 (36) |

such estimates of their life expectancy as hard-hitting, causing uncertainty and concerns. Some patients felt a need for immediate anti-tumor treatment after discussing such estimates, whereas others said such estimates made them reflect on the need for treatment. Patients with COPD often did not receive information about their life expectancy, which was in line with their belief that it is not possible to predict a specific life expectancy for this disease trajectory. A few patients (would have) liked to receive more information about the expected trajectory of COPD, especially towards the end of life. In general, patients agreed that the need for information about one's estimated life expectancy depends on individual preferences.

> *The oncologist was very direct like "Bear in mind that you will die from this". Although she could not specify a term, it is almost certain that I will die from the cancer. It really hit me when I came home and started talking about it. I thought to myself that I could die in five months, which is not even the expectation, but it can also occur in five or fifteen years. I know I am hoping against better judgment, but that is something that occupies my mind.* (**Interviewee 4, Cancer**)

**Patient's preferences.** Some patients, mainly those with cancer, had discussed or recorded their end-of-life preferences. Those preferences concerned medical treatment, continuation of work, or funeral arrangements. In general, patients considered those conversations as difficult and confronting, but they also experienced it as a relief to have informed their relatives and physicians about their preferences.

**Euthanasia.**   Some cancer patients had discussed the option of euthanasia with their relatives, before discussing it with their physicians, mostly their general practitioner.

### 3.2 Timing

For patients with cancer, their prognosis was mainly discussed when it became evident that their disease was incurable, or when (curatively aimed) treatment had to be stopped due to disease progression. Some patients experienced those conversations as hard-hitting or shocking, especially when they had experienced improvement in their functional status. Other patients felt that clear prognostic information after a period of waiting (and uncertainty) had somehow been relieving, despite the fact that it involved bad news.

Patients with COPD mainly received prognostic information at the moment they had been informed of their diagnosis of COPD, which had mostly occurred several years earlier. Some patients preferred not to discuss their prognosis again, unless their functional status would deteriorate or they would require hospitalization due to an acute exacerbation of COPD. However, others suggested that a hospitalization would not be the right moment to discuss their prognosis because they might be too ill at that time.

*The messages that had the largest impact concerned the diagnosis, the [survival] statistics, and that I have only six months left to live. This information was completely unexpected.* **(Interviewee 3, Cancer)**

*Hospitalization for a lung attack could be a reason to talk about my life expectancy, but it is not the right moment. It is very difficult for patients to talk during a lung attack.* **(Interviewee 7, COPD)**

### 3.3 Setting

*Place.* Patients preferred discussions about prognosis with their attending medical specialists to be held privately, for example in a private room at the ward. Some patients preferred to have discussions about prognosis with their general practitioner during a house visit, instead of at the general practitioner's office.

**Attendees.**   Generally, patients appreciated the presence of their relatives during discussions about prognosis, for instance because relatives can help understand information, ask additional questions, comfort the patient, or talk on the patient's behalf. A few patients regarded the presence of relatives as a burden, because they wanted to protect their relatives from bad news or because they felt responsible for their relatives' emotions.

*I have a very good relationship with my parents, and especially my mother wants to be present during conversations with the oncologist. What I find difficult is that both my parents are not in a good condition and actually need more care themselves. I sometimes worry that accompanying me to the hospital for conversations with the doctor is a burden for them.* **(Interviewee 1, Cancer)**

*My general practitioner sometimes visited me in my own home. I could then talk to her about everything, about the stress, everything.* **(Interviewee 10, COPD)**

### 3.4 Physician–patient relationship

In general, patients were pleased with the conversations they had had with their physicians regarding their prognosis and end of life. Patients mentioned that an established or long-term

physician-patient relationship could make those conversations easier. A few patients said that end-of-life discussions are easier with their general practitioner, whom they had known for a longer period, than with their medical specialist.

*The oncologist is a very nice doctor, very kind, and very gentle. She clarifies everything to me. She also gives me alternatives because she knows that I do quite a lot of research myself.* **(Interviewee 14, Cancer)**

Patients believed that a clear explanation by the physician about their illness trajectory or treatment (options) could be helpful. Some experienced bluntness, unpreparedness, or apathy of physicians, which made the conversation less pleasant. A few patients believed that younger physicians are more open to prognostic or end-of-life discussions than older physicians.

*That conversation was not that long ago, during our last appointment with that doctor before he retired. My husband was with me at that time. The doctor entered the room and while he was still holding the doorknob, he said, "Mrs. F., your situation really sucks." He was blunt. What he said really hurt me. That is something a doctor should never say to anyone.* **(Interviewee 8, COPD)**

## 3.5 Responsibilities for clinicians

**Medical specialists, general practitioners and other clinicians.**   Patients with cancer mentioned that the medical specialist is the designated person to initiate end-of-life discussions when they think it is appropriate, for example, when their health status significantly deteriorates. However, patients themselves want to decide on the extent or continuation of such discussions. Some patients with COPD, however, explicitly said that discussing patient' non-medical preferences for the end of life is not the task of the medical specialist, who they thought to be mainly responsible for discussing medical treatments. A vast majority of patients, mainly those with cancer, thought that the general practitioner has a primary role in end-of-life discussions and should at least offer the patient the opportunity to talk about his or her future. Additionally, a few patients would like to involve other clinicians, e.g. psychologists, because they might have more time than medical specialists.

*I do not think that medical specialists should talk about patients' future and expectations. I think specialists are busy helping people who are sick or short of breath.* **(Interviewee 10, COPD)**

*I have to admit that the conversations with the psychologist in the hospital really contributed to my positive attitude towards life. The conversations were really nice and straightforward. "You will have to start thinking about your future." "I have no future." "Well, tomorrow is also in the future, so is next week, and six weeks from now. Or do you think you will die this summer, and then I have to tell you that is not the expectation." The psychologist was very clear, which can be confronting for some people, but she really hit the right chord for me.* **(Interviewee 04, Cancer)**

## 3.6 Recommendations

All patients provided recommendations for physicians (Table 2). In summary, they suggested that physicians should assess the patient's need for prognostic information, deliver a clear

**Table 2. Recommendations for physicians for discussions about prognosis and end of life.**

**General**

- Greet with a handshake, stay calm and friendly
- Employ eye contact
- Choose an appropriate (i.e. private) place
- Reserve enough time for the conversation
- Be clear, to-the-point, and prepared
- Listen actively to the patient
- Encourage questions
- Check patient understanding
- Respond to the patient's emotions to the information
- Use appropriate language (i.e. avoid jargon)
- Check with the patient to have another person present (i.e. relatives)

**Specific**

- Initiate the conversation about disease course, expectations, and end of life
- Assess the patient's need for prognostic information
- Tailor the prognostic information to the patient
- Check with the patient and relatives about the patient's needs and preferences for end of life care
- Inform the patient about relevant and practical things to arrange in end of life.
- Check with the patient if a (new) treatment still fits in his/her vision for end of life
- Check with the patient about an advance directive
- Check with the patient to involve other clinicians in end-of-life discussions
- Repeat prognostic information in response to significant deterioration in the patient's condition or situation
- Direct the patient to trustful sources about the disease

message in understandable language, check the patient's understanding, show sensitive non-verbal communication, reserve sufficient time, and involve the patient's relatives or other clinicians.

# 4. Discussion and conclusion

## 4.1 Discussion

Patients prefer their medical specialist to initiate discussions about prognosis and end of life, but they want to negotiate the continuation and content. Those discussions are facilitated by the attendance of relatives, but can be hindered by high emotional or physical burden for relatives [15]. Patients prefer discussions regarding prognosis and end of life by physicians with whom they had an established and long-term relationship, by younger physicians who were more open to such discussions, and by physicians who delivered clear, sensitive communication (both verbal and non-verbal). Patients in our study seemed open to discuss their preferences with general practitioners and other clinicians (e.g. psychologists).

Patients with advanced cancer had had discussions about prognosis at clear-cut moments of deterioration (e.g. discontinuation of treatment due to disease progression), in contrast to patients with advanced COPD who were mostly informed about the prognosis at the moment of diagnosis. Generally, physicians may experience difficulties in identifying the moment of transition to of the end of life in patients with chronic organ failure, due to the unpredictable illness trajectory, which could make them more reluctant to initiate discussions about prognosis [16–21]. Furthermore, physicians have concerns of taking away patients' sense of hope by discussions about prognosis [21]. Although physicians prefer to initiate discussions about prognosis at clear-cut moments of deterioration, such as an acute exacerbation, our findings suggest that patients with COPD may disagree [21]. Patients may not be able to adequately communicate during such exacerbations, due to severe dyspnea or anxiety [22].

Patients with COPD thought that pulmonologists should only discuss patients' medical treatment, and refrain from discussing patients' non-medical end-of-life preferences. We

suggest three possible explanations. First, pulmonologists may tend to emphasize patients' symptoms more than other aspects, such as their social life, during consultations [23]. Furthermore, patients with COPD mainly have encounters with pulmonologists during an acute exacerbation, which may lead them to think that the pulmonologist is primarily responsible for medical treatment. Second, patients with advanced COPD seem less willing to have discussions about prognosis than patients with advanced cancer [24]. Unlike patients with advanced cancer, patients with advanced COPD may not realize to what extent their illness is life limiting [19]. Lastly, patients with advanced COPD may believe that end-of-life discussions do not contribute to their quality of life. A study by Duenk (2017) found that advance care planning discussions did not improve overall quality of life of patients with COPD hospitalized for an acute exacerbation [25].

There were several limitations to our study. First, patients were recruited by their treating medical specialists, which could have resulted in the inclusion of patients who had a good relationship with their physician, who felt obliged to participate or who were (overly) positive in their answers. However, patients were guaranteed confidentiality of their answers. In addition, two pulmonologists enrolled seven patients with COPD. Second, our sample had no diversity in patients' race or ethnicity, which has been found to influence preferences for prognostic communication [26]. Third, all interviews were conducted by one interviewer (CO), which supports a consistent approach to the interviews but could also create bias. However, potential bias was neutralized through independent analysis of the data by CO and LGGvL. Lastly, the testing of the interview guide with one person might not have been sufficient. However, the interview guide was frequently reevaluated with a representative of a patient and relative advisory board.

## 4.2 Conclusion

Patients with an advanced illness consider discussions about their prognosis and end of life important, but they want to negotiate the continuation or content of those discussions. Patients with cancer have those discussions at more clear-cut moments than patients with COPD, which may be due to the predictability of the illness trajectory. Patients with COPD did not consider the pulmonologist as the main responsible clinician to have end-of-life discussions about their non-medical preferences. Physicians are recommended to deliver a clear message in understandable language, check the patient's understanding, show sensitive non-verbal communication, reserve enough time, and involve the patient's relatives or other clinicians.

## 4.3 Practice implications

We recommend physicians to initiate discussions about prognosis, especially in response to significant deterioration of the patient's condition. They should assess patients' need for types of prognostic information, and for reflection on certain end-of-life topics. Prognostic or end-of-life discussions with patients with COPD may be more difficult due to their unpredictable illness trajectory. However, physicians should still address the illness course of COPD with patients and emphasize that COPD is life limiting. Future studies should assess patients' need for prognostic or end-of-life discussions, which could help physicians to tailor those discussions. Further research on physicians' own perceptions of such discussions for patients with different illness trajectories could unveil data on barriers or facilitators. The information on such studies and patients' recommendations for physicians in this study could be incorporated in education and communication skills training about prognostic or end-of-life discussions.

## Acknowledgments

The authors would like to thank all the patients who participated in this study and the medical oncologists and pulmonologists who recruited patients. We are very grateful to Liesbeth Smedinga for her input with the interview guide and for recommending someone for a trial interview.

## Author Contributions

**Conceptualization:** Catherine Owusuaa, Carin C. D. van der Rijt, Agnes van der Heide.

**Formal analysis:** Catherine Owusuaa, Liza G. G. van Lent.

**Funding acquisition:** Carin C. D. van der Rijt.

**Investigation:** Catherine Owusuaa.

**Methodology:** Catherine Owusuaa, Adriaan van 't Spijker, Carin C. D. van der Rijt, Agnes van der Heide.

**Writing – original draft:** Catherine Owusuaa.

**Writing – review & editing:** Liza G. G. van Lent, Adriaan van 't Spijker, Carin C. D. van der Rijt, Agnes van der Heide.

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
