## [Decision Letter · Decision Letter 0]

20 Jun 2022

PONE-D-22-07372Discussing prognosis and the end of life with patients with advanced cancer or COPD: An interview studyPLOS ONE

Dear Dr. Owusuaa,

Thank you for submitting your manuscript to PLOS ONE. After careful consideration, we feel that it has merit but does not fully meet PLOS ONE’s publication criteria as it currently stands. Therefore, we invite you to submit a revised version of the manuscript that addresses the points raised during the review process.

We look forward to receiving your revised manuscript.

Kind regards,

Felix Bongomin, MB ChB, MSc, MMed, FECMM

Academic Editor

PLOS ONE

Journal Requirements:

3. Please remove your figures from within your manuscript file, leaving only the individual TIFF/EPS image files, uploaded separately.  These will be automatically included in the reviewers’ PDF.

Additional Editor Comments:

Please consider the comments suggested by the reviewer and revise your manuscript accordingly.

Reviewers' comments:

Reviewer's Responses to Questions

**Comments to the Author**

1. Is the manuscript technically sound, and do the data support the conclusions?

Reviewer #1: Yes

2. Has the statistical analysis been performed appropriately and rigorously? 

Reviewer #1: N/A

3. Have the authors made all data underlying the findings in their manuscript fully available?

Reviewer #1: Yes

4. Is the manuscript presented in an intelligible fashion and written in standard English?

Reviewer #1: Yes

5. Review Comments to the Author

Reviewer #1: There are some grammatical errors

This is an important study as it will guide patient centered discussions at the end of life

Here are some comments

• The title: It may sound better to say a qualitative study than an interview study/

• The authors should look at COREQ checklist for reporting qualitative research to ensure all required items to be reported in qualitative research are reported

• Include a section in the manuscript on reflexivity.

• Please give a little more detail on the study setting are these referral settings/ tertiary settings or primary care settings. where were the interviews conducted, what was this location of their choice.

• The methodology used is not mentioned or theoretical framework for the study

• Please indicate the sampling method used

• Were field notes taken during interviews?

• In the First paragraph under results section the statement ”patients were slightly more female” does not sound grammatically correct, please improve this

• In the analysis It will be clearer to read if the subthemes are include in the body of the manuscript as subheadings

• Was any software used in analysis?

• Further depth in analysis would be helpful for example how did patient characteristics influence the discussions e.g where there patients characterised such as age , ginder etc tht were were observed with particular discussion e.g which type of patients discussed euthanasia and why

• Are there reasons why COPD patients got information on prognosis earlier at diagnosis and cancer one got it later

• How do the results of your study compare with previous studies.

6. PLOS authors have the option to publish the peer review history of their article (what does this mean?). If published, this will include your full peer review and any attached files.

Reviewer #1: No

---

## [Author Response · Author response to Decision Letter 0]

22 Aug 2022

Dear Chief Editor,

Thank you for the opportunity to revise our manuscript (PONE-D-22-07372). We appreciate the attentive reviews and constructive suggestions.

Please see below a point-by-point response to the reviewer’s comments in the first column with our responses in the second column, including how and where the text was adjusted. The textual changes made in the revised manuscript are marked using track changes.

We look forward to your response on our revised manuscript.

Sincerely,

On behalf of all the co-authors

Catherine Owusuaa 

Manuscript Title: Discussing prognosis at the end of life with patients with advanced cancer or COPD. An interview study

This is an important study as it will guide patient centered discussions at the end of life

 Here are some comments

 Thank you for your positive evaluation of our study.

• The title: It may sound better to say a qualitative study than an interview study/ 

- Thank you for the suggestion. We have made the suggested change.

• The authors should look at COREQ checklist for reporting qualitative research to ensure all required items to be reported in qualitative research are reported 

- We used the Standards for Reporting Qualitative Research (SRQR) guideline to structure the manuscript, as reported in paragraph 2.4 of the manuscript. The SRQR is similar to the suggested COREQ for reporting qualitative research. 

• Include a section in the manuscript on reflexivity. 

- We added a section on reflexivity in paragraph 2.4 of the manuscript. 

• Please give a little more detail on the study setting are these referral settings/ tertiary settings or primary care settings. where were the interviews conducted, what was this location of their choice. 

- Patients were included from one academic and two teaching hospitals. The interviews were conducted at a location of the patients’ choice, which was mainly at their houses.

 Details on the study are now provided in Table 1. We have added more clarification in the first paragraph of the results section.

• The methodology used is not mentioned or theoretical framework for the study 

- We added that we used a phenomenological interpretive approach in paragraph 2.4 of the manuscript. 

• Please indicate the sampling method used 

- Medical specialists in the participating hospitals consecutively selected patients from their outpatient clinics who probably had a life expectancy of on year or less. This has been clarified in paragraph 2.1 of the methods section.

• Were field notes taken during interviews? 

- Added paragraph 2.3: “The researcher took field notes during the interviews.”

• In the First paragraph under results section the statement ”patients were slightly more female” does not sound grammatically correct, please improve this 

- We have changed this is the concerned paragraph.

• It will be clearer to read if the subthemes are include in the body of the manuscript as subheadings 

- We have added the subthemes in italic at the start of the paragraphs throughout the manuscript.

• Was any software used in analysis? 

- No, we did not us specific software for the qualitative analysis. 

• Further depth in analysis would be helpful for example how did patient characteristics influence the discussions e.g where there patients characterised such as age , ginder etc tht were were observed with particular discussion e.g which type of patients discussed euthanasia and why

- We do not believe that our qualitative research (interview) approach allows us to analyse associations between characteristics of the participants and of the conversations.

• Are there reasons why COPD patients got information on prognosis earlier at diagnosis and cancer one got it later

- This is addressed in the second paragraph of the discussion section.

• How do the results of your study compare with previous studies. 

- We agree that this important. There were some differences between our study and previous studies, as already explained in the discussion section: The discussions about prognosis in our study were facilitated by the attendance of relatives. However, previous studies showed that discussions can be hindered by high emotional or physical burden for relatives. Previous studies also concluded that physicians prefer to initiate discussions about prognosis at clear-cut moments of deterioration, such as an acute exacerbation. Our findings suggest that patients with COPD may disagree.

---

## [Editor Report · Decision Letter 1]

24 Aug 2022

Discussing prognosis and the end of life with patients with advanced cancer or COPD: A qualitative study

PONE-D-22-07372R1

Dear Dr. Owusuaa,

We’re pleased to inform you that your manuscript has been judged scientifically suitable for publication and will be formally accepted for publication once it meets all outstanding technical requirements.

Kind regards,

Felix Bongomin, MB ChB, MSc, MMed, FECMM

Academic Editor

PLOS ONE
---

## [Editor Report · Acceptance letter]

30 Aug 2022

PONE-D-22-07372R1 

Discussing prognosis and the end of life with patients with advanced cancer or COPD: A qualitative study 

Dear Dr. Owusuaa:

I'm pleased to inform you that your manuscript has been deemed suitable for publication in PLOS ONE. Congratulations! Your manuscript is now with our production department. 

Kind regards, 

on behalf of

Dr. Felix Bongomin 

Academic Editor

PLOS ONE